# A Deep Learning Approach for Arabic Manuscripts Classification

**DOI:** 10.3390/s23198133

**Published:** 2023-09-28

**Authors:** Lutfieh S. Al-homed, Kamal M. Jambi, Hassanin M. Al-Barhamtoshy

**Affiliations:** 1Department of Computer Science, Faculty of Computing and Information Technology, King Abdulaziz University, Jeddah 21589, Saudi Arabia; kjambi@kau.edu.sa; 2Department of Information Technology, Faculty of Computing and Information Technology, King Abdulaziz University, Jeddah 21589, Saudi Arabia; hassanin@kau.edu.sa

**Keywords:** Arabic manuscript, deep learning, Region of Interest (RoI), Faster R-CNN, LSTM

## Abstract

For centuries, libraries worldwide have preserved ancient manuscripts due to their immense historical and cultural value. However, over time, both natural and human-made factors have led to the degradation of many ancient Arabic manuscripts, causing the loss of significant information, such as authorship, titles, or subjects, rendering them as unknown manuscripts. Although catalog cards attached to these manuscripts might contain some of the missing details, these cards have degraded significantly in quality over the decades within libraries. This paper presents a framework for identifying these unknown ancient Arabic manuscripts by processing the catalog cards associated with them. Given the challenges posed by the degradation of these cards, simple optical character recognition (OCR) is often insufficient. The proposed framework uses deep learning architecture to identify unknown manuscripts within a collection of ancient Arabic documents. This involves locating, extracting, and classifying the text from these catalog cards, along with implementing processes for region-of-interest identification, rotation correction, feature extraction, and classification. The results demonstrate the effectiveness of the proposed method, achieving an accuracy rate of 92.5%, compared to 83.5% with classical image classification and 81.5% with OCR alone.

## 1. Introduction

There are many scientific, medical, and religious disciplines where ancient Arabic manuscripts are valued as references. As such, the number of maintained ancient manuscripts in libraries worldwide is huge. According to Saabni and El-Sana [1], about 7 million manuscripts survived from 90 million manuscripts written between the 7th and 14th centuries. Unfortunately, natural forces, such as humidity, temperature, and air pollution, have severely damaged these manuscripts, which degrades their quality [2]. Accordingly, many of these manuscripts are missing parts, pages, or metadata [3]. The manuscripts called unknown (i.e., “majhoola” in Arabic) are those with missing identification information. Unknown manuscripts involve unknown authors, titles, publishing dates, and topics. Although these manuscripts are unknown, their values are significant [4]. But, huge efforts are required to extract information from these manuscripts, index them, and make them easily accessible and retrieved [5]. For unknown manuscripts, it is essential to identify meta-information and extract catalog-related information, such as (1) author, (2) language, (3) subject, (4) topics, and (5) publishing date [6], which is used for indexing purposes in libraries. However, given the huge number and degraded quality of such manuscripts, manual extraction of information from the manuscript is challenging [7].

Catalog cards contain information about books and manuscripts maintained in libraries. For the ancient manuscripts, these cards represent the diligent work of librarians and researchers in identifying these invaluable resources, facilitating easy access to the information enclosed within these resources. Catalog cards for Arabic manuscripts typically include essential parameters, such as the title, the author’s name, the manuscript’s beginning and end, and the number of images [4]. Additional technical parameters, like the number of lines, number of words, page size, paper type, and other details, may be inferred, but they are not considered essential. These technical parameters serve various purposes, such as identifying the era in which the manuscript was written, reconstructing fragmented manuscripts, determining page layouts, and even discerning the author’s style in cases where the author’s name is unknown. Additionally, they can provide insights into the scribe’s identity and the copying date when such information is missing [8]. Furthermore, the information provided for these parameters can be used to automatically process the ancient manuscripts and extract apparent and hidden information. Yet, there are many problems facing processing these catalog cards, such as using handwriting to fill in these cards, having multiple forms and shapes for the cards, and providing scanned copies of these cards with different qualities, rotations, resolutions, etc. Accordingly, these cards cannot be processed using a standard layout processing technique.

To extract information from documents, deep learning is employed. Deep learning algorithms performed well in the image processing domain, including character and text recognition. Conventional Neural Networks (CNNs) are specifically used for image processing and computer vision tasks. While they perform well in Optical Character Recognition (OCR) of typed text, they struggle to capture multiple objects within the image. The main drawback of CNNs is their approach of processing the entire image as a single object, making it challenging to capture multi-object images. To address this limitation, various techniques, such as Region-based CNN (R-CNN), Fast R-CNN, Faster R-CNN, and Mask R-CNN, have been proposed. Faster R-CNN, in particular, is a deep neural network algorithm that efficiently detects and recognizes objects in the image compared to R-CNN. This algorithm can be efficiently used with proper preprocessing steps with complex OCR applications due to its high performance in complex object detection and recognition tasks [9]. On the other hand, Long Short-Term Memory (LSTM) networks are well suited for learning patterns from sequences of data, making them a suitable choice for OCR applications [10].

Information extraction from the catalog cards, especially for unknown ancient Arabic manuscripts, is significant yet challenging due to the low quality of the cards, the variation of the layout of cards across different libraries, and variations in the writers’ styles. Nevertheless, the advances in deep learning allow for processing complex and low-quality images for the target documents. These advances serve as the motivation for this research. Accordingly, this research aims to use deep learning to automatically extract information from the catalog cards, specifically identifying whether the manuscript is known or unknown.

This paper proposes a generalized framework for processing catalog cards of ancient Arabic manuscripts. The proposed approach identifies the parameters and their values using a set of processing steps and a deep learning technique, specifically, Faster R-CNN and LSTM. The extraction process is template- and layout-free, making it suitable for various card formats in different libraries. As such, the contributions of this paper are as follows: (1) Building a complete framework for ancient Arabic manuscripts processing based on the catalog cards with parameter identification. The processing aims to classify the cards into known and unknown, a crucial step in extracting information from ancient Arabic manuscripts. (2) Using Faster R-CNN to locate and classify the Arabic text into typed and handwritten text. The deep learning architecture facilitates accurately spotting regions representing typed and written text. (3) Using Faster R-CNN to recognize the parameters and parameters’ values in the cards. This recognition extracts parts of the significant information found in the catalog cards of the ancient Arabic manuscripts.

## 2. Literature Review

Various approaches have been proposed for processing and extracting knowledge from ancient manuscripts in different languages, including Arabic manuscripts [11]. The processing of such manuscripts depends on language-related algorithms, methods, and applications. While natural language processing (NLP) methods share much in common, some variations and challenges distinguish the Arabic-related methods and applications. Generally, manuscript processing requires image enhancement and segmentation into words and characters. Segmentation and recognition can be on the word or character levels. The segmentation is then followed by feature extraction, and a supervised model is trained to classify words/characters. Yet, there is a segmentation-free approach for document and manuscript processing. The segmentation-free approach does not implement any segmentation. Instead, features are extracted from fixed or sliding windows [12].

The literature on ancient and historical manuscript processing is not as extensive as that on modern document processing. However, processing historical documents can benefit from the advances in modern document processing. Yet, ancient and historical manuscript processing poses more challenges due to the poor quality of the images [2,13,14,15]. Moreover, tasks associated with the ancient manuscripts are not typically required for modern documents, such as author identification and manuscript dating. Their applications have not been extensively investigated in the literature [15]. Nevertheless, many of these tasks depend mainly on OCR-related applications, with specific preprocessing or post-processing stages.

Early Arabic OCR approaches are segmentation based, with variations in the image processing techniques. Alma’adeed, et al. [16] proposed a technique based on image normalization, segmentation, skeletonization, and edge-based feature extraction. The words are extracted, and then each word is compared to a dictionary of words for similarity matching using image processing steps. The results over a dataset of 1000 samples achieved an accuracy of 45%. Schambach, et al. [17] proposed a technique for recognizing Arabic words using a sliding window and feature extraction. The preprocessing steps depend on extracting and smoothing the contour of the characters. The problem is converted into character recognition as the sliding window is applied. The experiments were conducted on the International Conference on Document Analysis and Recognition (ICDAR)-2007 competitions dataset for word recognition, with an accuracy of 94.85%. AlKhateeb, et al. [18] proposed Arabic-OCR based on image enhancement, noise removal, word segmentation, intensity features extraction, and the Hidden Markov Model (HMM) classifier. Applying this technique to the ‘Institut fur Nachrichtentechnik’ and ‘Ecole Nationale d’Ingénieurs de Tunis’ (IFN/ENIT) dataset [19], which involves 32,492 handwritten words, achieved an accuracy of 95.15%. Likforman-Sulem, et al. [20] proposed sliding window-based feature extractions without implementing any segmentation on the input text. Statistical and structural features are extracted from each window, and the HMM classifier is implemented. Applying this technique to the IFN/ENIT dataset and evaluating the results on 6033 words resulted in an accuracy of 89.42%. Generally, early approaches use preprocessing steps to ease the expected segmentation errors. A comprehensive survey of the segmentation-based OCR and the existing segmentation techniques are presented by Ali and Suresha [21].

Recent trends in OCR techniques development have shifted towards segmentation-free approaches. These advancements have revealed that the use of segmentation techniques often leads to significant or trivial errors that can degrade OCR performance. In segmentation-free techniques, words are recognized without the need for a prior segmentation step. For instance, Nemouchi, Meslati [13] proposed an OCR approach to recognizing handwritten city names. The process includes binarization, smoothing and skeletonizing, and feature extraction, including moments, Fourier descriptors, and Freeman’s chain code. Multiple classifiers, including Neural Networks (NNs), K-Nearest Neighborhood (KNN), and Fuzzy C-Means (FCM), are implemented for recognition purposes. The experiments, conducted on 270 images of city names, achieved an accuracy of 70%. Sabbour and Shafait [22] proposed an OCR that involves normalizing image sizes, segmenting pages into lines, performing ligature segmentation, and utilizing shape-based feature extraction. This approach yielded an accuracy rate of 86%. Nashwan, et al. [23] proposed a computationally efficient holistic Arabic OCR using clustering and classification algorithms. Initially, input images are preprocessed and segmented into lines and words. Then, a lexicon reduction technique, based on clustering similar words, was utilized to reduce word recognition time. In the training phase, words are extracted, and the clustered machine learning algorithm is trained. Applying this technique to datasets of single-word images containing 1152 words resulted in an accuracy of 84%. A comprehensive survey of the segmentation-free OCR techniques is presented by Sabbour and Shafait [22].

Generally, the challenges of segmentation-free techniques include the training and testing time, which have been alleviated with the advances in hardware technology. Conversely, the advantage of the segmentation-free approach is the achieved accuracy compared to the segmentation-based techniques [24].

Overall, various deep learning algorithms have been utilized in recent OCR techniques to improve recognition accuracy. CNNs are a specific type of NN architecture designed for processing pixel data and have found applications in various computer vision tasks, including feature extraction, enhancement, object detection, and recognition. CNNs have become one of the core techniques in image processing due to their accuracy and ability to process complex data for complex tasks. A CNN typically consists of several basic layers organized in a sequence of processing units. These layers include the convolutional layer, which uses filters and kernels for abstracting the input image as a feature map. The pooling layer is used to down-sample the feature map, and the fully connected layer is used for prediction. The structure of a CNN can be customized and enriched, based on the type of application and the input form [25].

For Arabic OCR, El-Sawy, et al. [26] developed a CNN-based OCR system. The developed system did not apply any processing, correction, or segmentations to the input images of individual characters. Generally, CNNs do not require explicit feature extraction; instead, features are automatically learned and filtered while training the CNN. The CNN architecture includes an input layer, two convolutional layers, two max pooling layers, a fully connected layer, and an output layer. The network was trained on a dataset of 28 distinct characters, without considering the variation based on their position within words. The dataset consisted of 16,800 input images written by 60 participants of different ages, gender, etc. The results showed that the accuracy of the developed approach is 94.9%.

El-Melegy, et al. [27] developed another CNN-based Arabic OCR system designed for handwritten text recognition, specifically for recognizing literal amounts in Arabic texts. The developed network consists of an input layer; three layers for each of the following—convolutional, batch normalization, rectified linear unit (ReLU), activation, max pooling, and fully connected layers; and an output layer. The experiments were conducted on the Arabic Handwriting Data Base (AHDB) [28] dataset, which consists of 4971 samples categorized into 50 classes, each representing a single literal amount, with 100 samples per class. The results showed that the accuracy of the developed approach was 97.8%. Another CNN-based OCR system was developed by Hamida, et al. [29], which consists of an input layer; two layers of each of the following—convolutional (20 filters of 595 size), pooling (494 sizes) using rectified linear unit (ReLU), and fully connected; and an output softmax layer. The experiments were conducted using the National Institute of Standards and Technology (NIST) special dataset of digits, called Modified NIST (MNIST) [30], consisting of handwritten images of 240,000 samples. The results showed that the accuracy of the developed approach was 99.22% for characters and 99.74% for digits.

Several other CNN-based approaches for Arabic OCR have been developed. Neri, et al. [31] designed a CNN consisting of an input layer, three convolutional layers, two ReLU max pooling layers, a fully connected layer, and one softmax output layer. The experiments were conducted on two customized datasets of digits and symbols, each comprising 1193 images, achieving an accuracy of 98.19%. Younis [32] developed a CNN with batch normalization to prevent overfitting. The experiments were conducted on the AlexU Isolated Alphabet (AIA9K) [33] and the Arabic Handwritten Character Dataset (AHCD) [16] datasets, resulting in accuracies of 94.8% and 97.6%, respectively. Ashiquzzaman, et al. [34] developed a CNN with augmentation and dropout processes. The developed network consists of an input layer, four convolutional layers, one max pooling layer, a size-dropout layer, three fully connected layers, and one output layer. The results achieved a 99.4% accuracy on the Center for Microprocessor Applications for Training Education and Research (CMATER) dataset. Balaha, et al. [35] proposed dual CNN models built by 10 layers, with optimization and dropout techniques for character recognition. The proposed technique was evaluated based on three datasets, one of which was developed, delivered, and described, along with the technique with AIA9k [33] and CMATER [36].

To ease the OCR process and enhance its performance in documents containing multiple lines, words, and regions, word-spotting techniques have been proposed. Word-spotting techniques are developed to locate and recognize specific words or phrases within images of documents. Word spotting is typically implemented as a pre-processing step before OCR to extract specific information among various information provided. Daraee, et al. [37] proposed a framework for word spotting for the applications of query-by-example and query-by-string. A Monte Carlo dropout CNN is used for the word-spotting task. Monteiro, et al. [38], on the other hand, proposed a framework combining object detection using the YOLO technique, which uses CNN deep learning architecture, with OCR. This integrated framework allows for the recognition of both textual and non-textual objects, particularly in printed stickers. Jemni, et al. [39] proposed a keyword-spotting framework that relies on a transformer-based deep learning architecture, eliminating the need for CNN layers. This approach aims to generate more robust and semantically meaningful features, thereby enhancing keyword-spotting performance.

The main limitation of CNNs is their inability to recognize multiple objects in images, which is commonly the scenario in document images. Accordingly, R-CNN was proposed by Girshick, et al. [40] to overcome the drawback of CNN. R-CNN works by extracting approximately 2000 region proposals from each image and filtering them into the Regions of Interest (ROIs). However, R-CNN is slow and time inefficient. Fast R-CNN was also proposed by [41] to overcome the limitation of the R-CNN. In the Fast R-CNN, the input image is fed into the CNN, rather than the region proposals. Region proposals are extracted from the convolutional feature map, warped into squares, and then a RoI pooling layer is used to restructure them into a fixed size, so they can be fed into a fully connected layer. The problem with Fast R-CNN is the utilization of the selective searching algorithm on the feature map by which the region proposals are identified. In Faster R-CNN [42], another neural network called Region Proposal Network (RPN) is used to locate the RoIs in the image. Mask R-CNN [43], an improved object detection model based on Faster R-CNN, has an impressive performance on various object detection and segmentation benchmarks, such as COCO challenges [44] and the Cityscapes dataset [45]. Mask R-CNN generates a mask for each Region of Interest (RoI), which can fulfill both detection and segmentation tasks. However, segmentation may not be required in many OCR applications. A summary of the related work is presented in Table 1. In summary, advanced deep learning algorithms are typically designed for recognizing single entities, such as characters, symbols, numbers, words, or phrases. Traditional OCR systems often struggle with handwritten text. To address these limitations, this paper proposes a framework for recognizing multiple entities to facilitate information extraction.

## 3. Methodology and Design

The Arabic catalog cards commonly involve various parameters organized in different layouts, as illustrated in Figure 1. These parameters provide significant information about the manuscripts, such as the author’s name, the writer’s name, the title, the topic, and others. The author’s name is the subject matter for the proposed approach, yet other information can also be extracted similarly.

The proposed framework identifies relevant text regions on the card among all other regions. Then, it classifies RoIs as typed and handwritten. A part of the handwritten text corresponding to the author’s name is analyzed, using image features to avoid the utilization of the OCR. A preliminary study of using OCR on such low-quality images led to poor results using the state-of-the-art techniques. The flowchart of the proposed methodology is illustrated in Figure 2. An alternative framework experimented with in this paper is also presented in Figure 3. Figure 4 and Figure 5 represent the baselines against which the proposed framework will be compared.

The alternative framework experimented on in the preliminary phase of the research shares common characteristics with the proposed framework, but uses the OCR, as illustrated in Figure 3. Unfortunately, as will be discussed, state-of-the-art Arabic OCR could not correctly recognize the handwritten text. The last two frameworks represent baseline models for comparison purposes; these are image-based and OCR-based, as illustrated in Figure 4 and Figure 5. The processes in these frameworks will be discussed accordingly.

### 3.1. Detection of the Text Regions and Text Type

After resizing, the input images representing the cards are processed using the Faster R-CNN algorithm. The Faster R-CNN processed the input image, regardless of its conditions and direction, and recognized two object types: handwritten and typed text. As a deep supervised learning algorithm, the Faster R-CNN is implemented in both the learning and prediction phases, as illustrated in Figure 6.

In the learning phase, an annotated dataset is used to train the constructed deep network. The training images are passed through a pre-trained CNN to extract high-level features from the image. ResNet was selected as the CNN backbone to implement the feature extraction step. Among the existing pre-trained CNN architectures, such as the VGG and MobileNet, ResNet was chosen for its high performance in object detection. The utilized ResNet50, an extension of ResNet34, consists of 50 layers. The first layer is a 7 × 7 convolution with 64 kernels. The following layers include a max pooling layer and 9 other 1 × 1 and 3 × 3 convolutions layers of 64 and 256 kernels. The remaining layers have more kernels, all feeding into the average pooling layer. The extracted features are then used to train the RPN, another convolution network that captures object regions in the images. The training of the RPN depends on the extracted features and the associated bounding box for the objects in the training images. Finally, a fully connected network is used for classifying and regressing the proposed regions and their bounding boxes. This network consists of two fully connected layers: one for classification and the other for bounding box regression.

In the prediction phase, the unannotated images are passed through the same pre-trained ResNet network to extract high-level features. The trained RPN is then used to generate region proposals with bounding boxes for the objects in the image. RPN slides a window (typically 3 × 3) over the feature map generated by the CNN and predicts a score and an adjustment of the bounding box for each location. The identified regions are then reshaped and aligned to a fixed size using RoIs-pooling in the RPN network. The fully connected layer is then used for both classifying the regions and refining the bounding boxes. The predicted objects undergo Non-Maximum Suppression (NMS) to eliminate redundant and non-confident objects.

### 3.2. Rotation Correction

The input images also passed through a rotation correction process to identify any rotation angle present. The implemented rotation correction depends on extraction, processing, and classification of connected components. These components are processed independently, and their orientations are identified individually. The overall orientation is determined based on the orientation with the highest confidence score. The image identifies the components as characters, separated by white space. In this step, the accuracy of the character segmentation is not highly significant for the results. Consequently, the framework uses white space-based segmentation and wrapping components with bounding boxes using the International Components for Unicode (ICU) layout engine.

### 3.3. Pair-Text Matching

In the proposed framework, a simple yet general layout analysis is implemented to match the parameter titles with their handwritten values, forming a set of parameter-value pairs. This step depends on the rotation angle found during the rotation correction process and the coordinates of each RoI in the image, as illustrated in Figure 7. The coordinates of each RoI are adjusted based on the identified angle, and new coordinates for these regions are stored for use In the matching process. First, the top-right corner of each RoI containing typed text is identified, taking into account that Arabic text is written from right to left. The sets of corners for all RoIs containing typed text are then sorted, primarily based on their y-coordinates and x-coordinates. Similarly, the RoIs containing handwritten text are sorted using the same criteria. Then, correspondence is established by merging the two sorted lists, ensuring that each handwritten RoI corresponds to an adjacent typed RoI.

### 3.4. Arabic Object Character Recognition

The recognized objects, along with their associated classes, are cropped and rotated based on the preceding steps. These manipulated images are then used as input for the next processing step, the OCR. The preprocessed input images undergo a recognition process that implements LSTM, another deep learning approach based on Recurrent Neural Network (RNN). LSTM is capable of processing data sequences, such as image pixels. The LSTM has cells that adjust their states iteratively during the learning process, enabling the capture of periodic patterns in the input sequences. As a result, it proves to be effective for Arabic OCR. This technique has been used for both printed and handwritten text within the proposed framework.

### 3.5. Feature Extraction

Named Entity Recognition (NER) is used to recognize and extract named entities in the values of the “name” parameter. NER assigns an identification to each input word in the text. The results of NER are used to populate the values for nine features used in the feature extraction step. These features correspond to the possible output of NER, which is related to four entity types: person, location, organization, and Miscellaneous. Each feature represents the beginning or continuation of these entities, while the ninth feature denotes a non-named entity. Although NER produces an output for each word, a frequency vector represents each card in the proposed framework. The features represented in the feature vector are illustrated in Figure 8.

### 3.6. Classification

The final step is to train and test the classification algorithms utilized. Various classifiers were employed, including the Linear Support Vector Machine (SVM), KNN (k = 3), Decision Tree (DT), Random Forest (RF), Artificial Neural Network (ANN), and Deep Neural Network (Deep-NN). The classification process in the proposed frameworks is implemented in two stages. The first stage aims to recognize the sub-image containing the “author name” parameter among all other identified text regions. The second stage classifies the region corresponding to the “author name” parameter value after pair-matching into known or unknown. In the alternative framework, classification is implemented for the text value identified with the region corresponding to the “author name” value. In the baseline image-based approach, classification is implemented on the entire input image after preprocessing (resizing and thresholding). Finally, in the OCR-based framework, classification is implemented in the vector space of the words extracted from the image dataset.

## 4. Experiments

The experiments were conducted using a collection of catalog cards corresponding to both known and unknown ancient Arabic manuscripts. The dataset was created and annotated with the true class labels for the objects under consideration, which include both handwritten and typed text. In this context, the typed text represented the parameters, and the handwritten text represented the values assigned to these parameters. Consequently, if the manuscript’s parameter values were either not provided or were denoted as ‘unknown’, the manuscript was classified as ‘unknown’. On the contrary, it was classified as known if the parameters were complete. The experiments were then conducted in the Python 3 programming language. Further discussion on the dataset and the experiments are provided in this section.

### 4.1. Dataset

A set of known and unknown ancient Arabic manuscripts and their associated cards were collected from various libraries. A total of 100 known manuscripts containing 8478 images and their associated cards and 100 unknown manuscripts containing 4732 images and their cards were collected. These manuscripts and their cards were collected from 10 different libraries. The catalog card contains the parameters, such as the category, name, title, etc., as shown in the empty card example in Figure 9. An example of the known manuscript and its associated card are presented in Figure 10, and an unknown manuscript and its card are presented in Figure 11.

### 4.2. Annotations

In the annotation step, the Computer Vision Annotation Tool platform (CVAT v2.7.1) [46] has been used to annotate the images, which will be used to train the region detector. The images are annotated with two object types, printed and handwritten texts, and the regions of interest and their boundaries are identified in the annotation process. Examples of known and unknown catalog cards are shown in Figure 12.

### 4.3. Implementation

The proposed framework was implemented using Python in the Anaconda Integrated Development Environment (IDE) and based on a set of libraries, as shown in Table 2.

### 4.4. Parameter Setting

As for the parameter settings, the experiments were conducted by splitting the dataset into tenfold, and the results were reported for the 200 images. The Faster R-CNN was trained with 150 iterations for each fold. The classification algorithms used are established using the following settings: For the KNN, k of 3 was utilized. A linear kernel was used for the SVM, a single layer for the ANN, and three (200, 100, 50) layers were used for the deep learning.

## 5. Results and Analysis

The output results of the proposed framework were evaluated using accuracy, recall, precision, and F-measure. These commonly utilized classification evaluators are calculated based on the outputs’ truePositive, trueNegative, falsePositive, and falseNegative portions, as presented in Equations (1)–(4).
(1)$$accuracy=\frac{truePositive+trueNegative}{truePositive+trueNegative+falsePositive+falseNegative}$$
(2)$$precision=\frac{trueNegative}{truePositive+falsePositive}$$
(3)$$recall=\frac{truePositive}{truePositive+falseNegative}$$
(4)$$F-measure=\frac{2\times precision\times recall}{precision+recall}$$

The results according to these measures are presented in Table 3. As noted, the proposed framework obtained higher accuracy, precision, and recall than the naïve OCR-based and image classification-based approaches. Accordingly, it can be concluded that using the parameters in the catalog cards with the semantic NER process performs far better than the blindly image-based classification and the imprecise OCR-based approach.

As given, the proposed framework outperformed the baselines and the alternative frameworks regarding accuracy, precision, recall, and f-measure. During the experiments, it was found that the tailored Faster R-CNN is very robust for the associated task. The Faster R-CNN achieved full results distinguishing the typed text from the written text, as shown in Table 4. However, some RoIs were not recognized, especially for the written text. Accordingly, for 3200 regions for typed text, 3104 were recognized fully or partially. As for the written text, for 2105 regions, 1979 regions were recognized. As for the bounding box identifying handwritten and typed text, the results were not as accurate as the classification, as listed in Table 5. Various regions were split into two regions, and some were combined for the written text. Figure 13 illustrates the resulting regions after running the Faster R-CNN processes.

Although the Faster R-CNN obtains satisfactory results in the proposed and the alternative frameworks, the OCR drops the results significantly in the alternative framework. Mostly, all the written texts inputted to the OCR were wrongly recognized. The correctly classified images in the alternative framework are not due to correctly recognizing the text; instead, the cards of the unknown manuscripts commonly have no value in the corresponding region for the author’s name. That is how the classifiers could recognize the unknown manuscripts, in which the vector of the unknown cards is commonly empty. In contrast, the vector of the known manuscripts has some input (although it is wrong).

Surprisingly, the results of the OCR have improved significantly in the baseline framework, as the images were processed as a whole instead of processing each sub-image solely. Still, the results of the OCR with the whole image are not accurate, but it was able to obtain some results with some classifiers, especially the DT. Overall, using the deep learning and classification approach, it was able to classify images into unknown and unknown types. Avoiding OCR was useful, as the written text was very random, and the image quality was bad.

## 6. Conclusions and Future Work

This paper proposed a framework for classifying ancient Arabic manuscripts into known and unknown categories. The framework primary focus is identifying parameters and recognizing their values from catalog cards associated with the manuscripts. The framework processes the input cards through several steps, including a region detection phase, rotation correction, and region alignments, followed by the final classification step. For this purpose, Faster R-CNN and various classification approaches were utilized. A dataset was collected from various libraries and annotated with bounding boxes corresponding to the objects of interest. The experiments conducted on this dataset showed that the proposed approach outperformed the classical approaches by 10% in terms of accuracy. The results showed that the proposed framework is very robust for detection and recognition tasks. The Faster R-CNN achieved full results distinguishing the typed text from the written text. However, some RoIs were not recognized, especially for the written text. Accordingly, for 3200 regions for typed text, 3104 were recognized fully or partially. As for the written text, for 2105 regions, 1979 regions were recognized.

Future work will be dedicated to processing all parameters found in the catalog cards and extracting information from these parameter values. Notably, the results were obtained without implementing any preprocessing steps, except for the resizing step. Therefore, in future work, we plan to incorporate preprocessing stages, including image enhancement and noise removal, as well as post-processing methods, such as context-based OCR improvement, to address instances where Regions of Interest (RoIs) were not accurately detected.

## Figures and Tables

**Figure 1 sensors-23-08133-f001:**
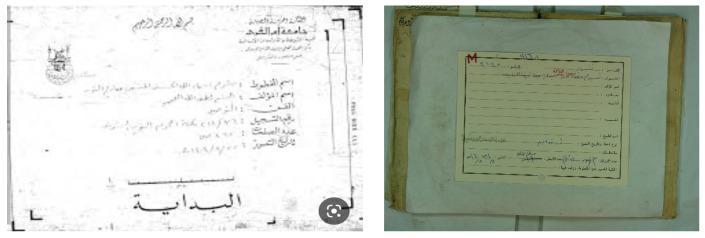
Example Catalog Cards with Different Parameters and Layout.

**Figure 2 sensors-23-08133-f002:**
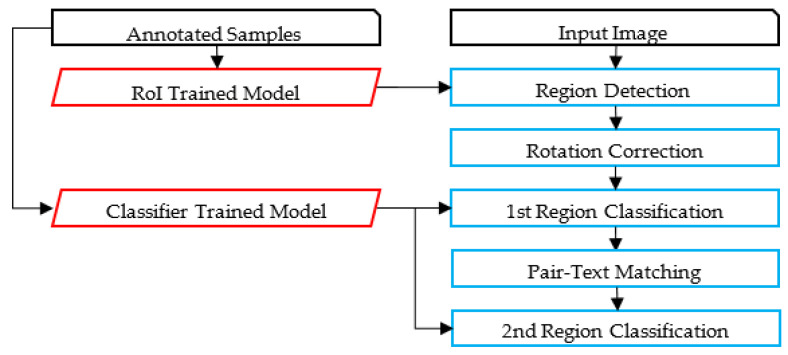
The Proposed Framework.

**Figure 3 sensors-23-08133-f003:**
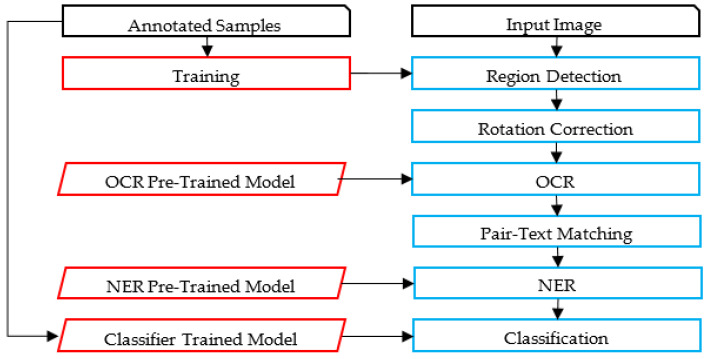
The Alternative Framework.

**Figure 4 sensors-23-08133-f004:**
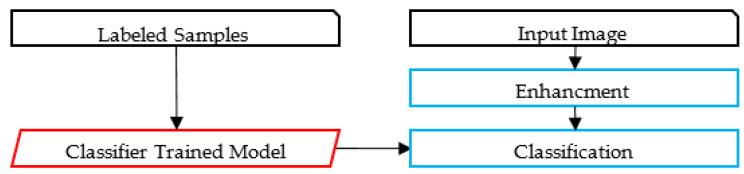
The Image-based Framework.

**Figure 5 sensors-23-08133-f005:**
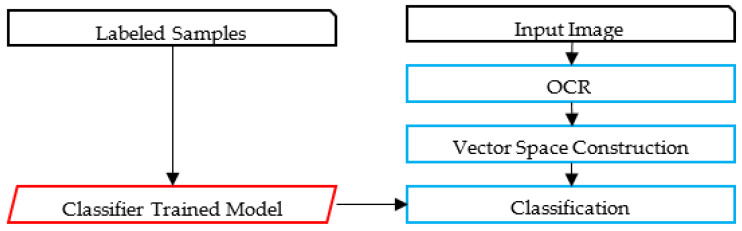
The OCR-based Framework.

**Figure 6 sensors-23-08133-f006:**
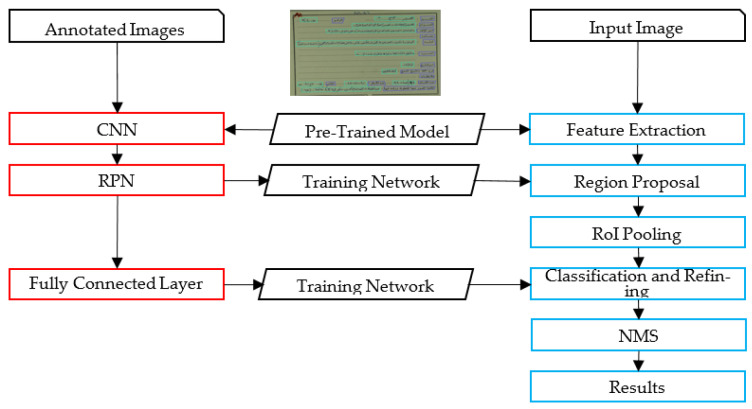
Faster R-CNN for Manuscript Card Processing.

**Figure 7 sensors-23-08133-f007:**
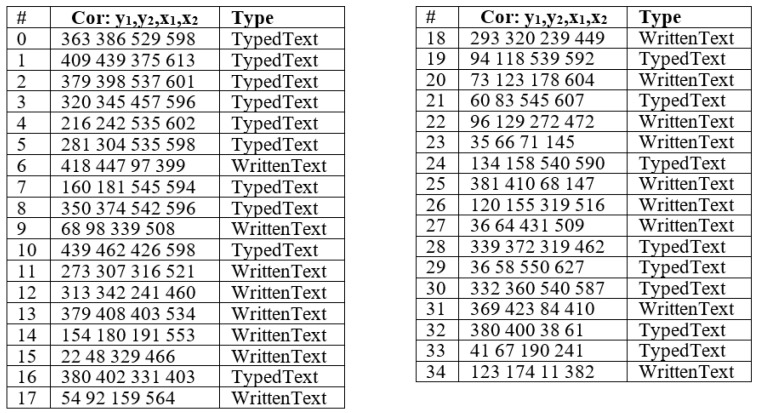
Example of the RoI for a Single Image.

**Figure 8 sensors-23-08133-f008:**
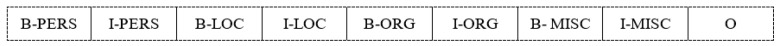
Feature Vector for Each RoI using NER.

**Figure 9 sensors-23-08133-f009:**
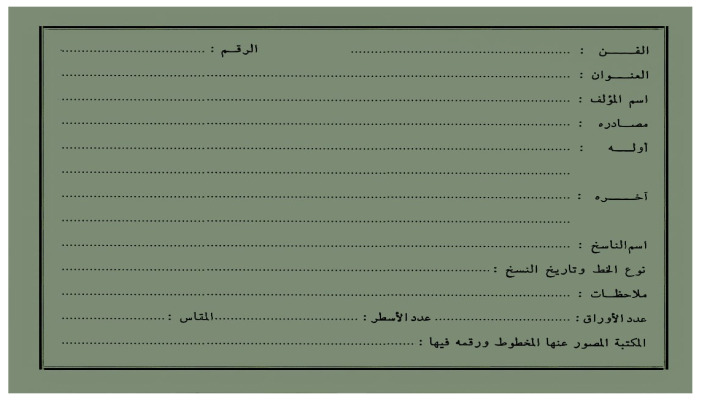
Catalog Card Template.

**Figure 10 sensors-23-08133-f010:**
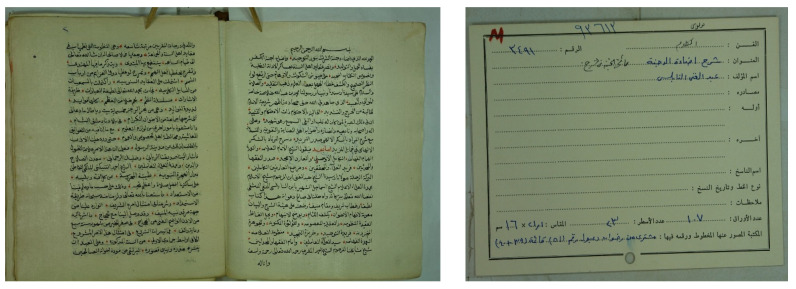
A Single Slice of a Known Manuscript and the Associated Card.

**Figure 11 sensors-23-08133-f011:**
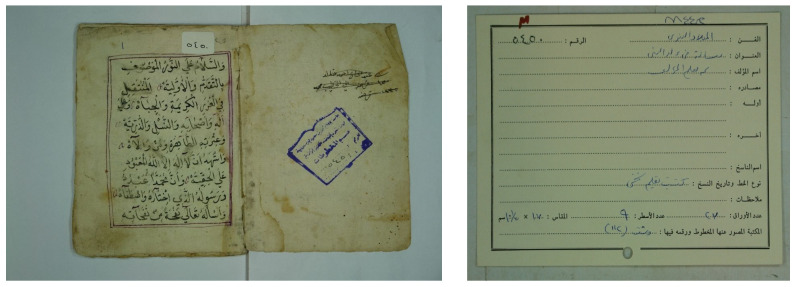
A Single Slice of an Unknown Manuscript and the Associated Card.

**Figure 12 sensors-23-08133-f012:**
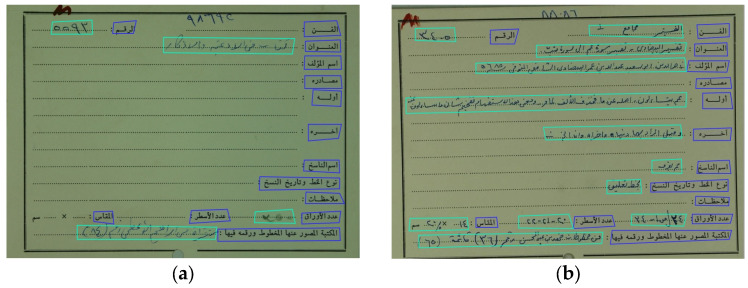
Example of Catalogue Cards: (**a**) Unknown Manuscript (**b**) Known Manuscript 4.3. Implementation.

**Figure 13 sensors-23-08133-f013:**
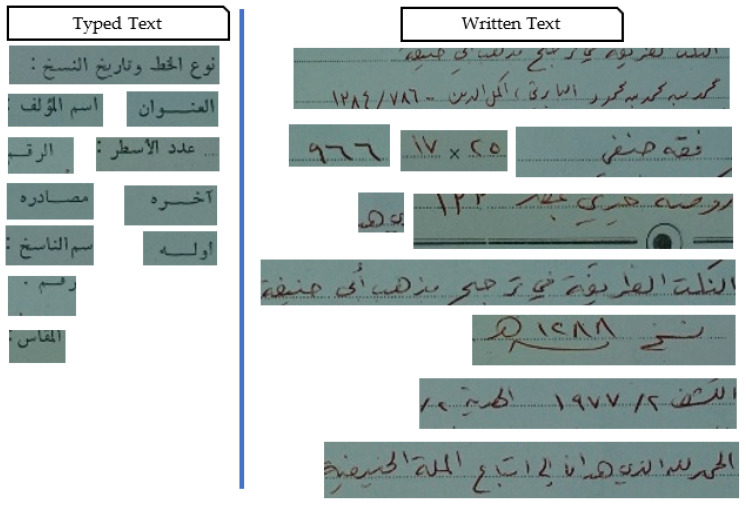
The Regions Resulted Using the Faster R-CNN.

**Table 1 sensors-23-08133-t001:** Summary of the Related Work.

Ref.	App.	Advantage	Disadvantage
[16]	Arabic OCR	Easy to interpret	Low performance
[17]		Noise tolerance	Not suitable for handwritten text
[18]		Fast and efficient	Not suitable for distorted text or noisy images
[20,22,23]		Noise tolerance	Not suitable for handwritten text
[26,27,29,31,34,35]		No preprocessing	Suits single-character images only
[37,38,39]	Spotting	No preprocessing	Captured single word only

**Table 2 sensors-23-08133-t002:** List of Utilized Libraries.

Library	Description
Numpy	Used for array manipulation, specifically for saving the titles and the information of the images and the output of region detection and to sort the regions based on the coordinates.
Scikit-Learn	Used for the classification step to implement the utilized classifiers.
OpenCV	Used for image reading, writing, and in the preprocessing stages with the OCR.
Tesseract 4.0	Used as the OCR engine based on LSTM with a pre-trained Arabic model.

**Table 3 sensors-23-08133-t003:** Results of the Unknown Manuscripts Classification.

KNN				
	Accuracy	Precision	Recall	F-Measure
Proposed Framework	87.0%	0.849	0.90	0.874
Alternative	82.0%	0.802	0.85	0.825
Image-based	79.0%	0.784	0.80	0.792
OCR-based	76.0%	0.755	0.77	0.762
SVM				
	Accuracy	Precision	Recall	F-Measure
Proposed Framework	88.0%	0.858	0.91	0.883
Alternative	86.0%	0.833	0.90	0.865
Image-based	80.5%	0.785	0.84	0.811
OCR-based	80.5%	0.785	0.84	0.811
ANN				
	Accuracy	Precision	Recall	F-Measure
Proposed Framework	84.5%	0.822	0.88	0.85
Alternative	79.5%	0.781	0.82	0.80
Image-based	73.0%	0.713	0.77	0.74
OCR-based	70.0%	0.696	0.71	0.702
Deep-NN				
	Accuracy	Precision	Recall	F-Measure
Proposed Framework	87.0%	0.849	0.90	0.873
Alternative	82.0%	0.802	0.85	0.823
Image-based	79.0%	0.784	0.80	0.792
OCR-based	76.0%	0.755	0.77	0.762
DT				
	Accuracy	Precision	Recall	F-Measure
Proposed Framework	92.5%	0.912	0.94	0.926
Alternative	90.0%	0.892	0.91	0.9
Image-based	82.5%	0.802	0.85	0.825
OCR-based	81.5%	0.789	0.86	0.823
RF				
	Accuracy	Precision	Recall	F-Measure
Proposed Framework	82.0%	0.802	0.85	0.825
Alternative	77.5%	0.772	0.78	0.746
Image-based	77.5%	0.772	0.78	0.746
OCR-based	71.0%	0.700	0.75	0.724

**Table 4 sensors-23-08133-t004:** Faster R-CNN Classification Results.

	Precision	Recall
Typed Text	1.0	0.97
Written Text	1.0	0.94

**Table 5 sensors-23-08133-t005:** Faster R-CNN Bounding Box Results.

	Splitted	Merged
Typed Text	0.0	0.0
Written Text	0.314	0.475

## Data Availability

Part of the data is available upon request.

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
