# Peer review of "A Deep Learning Approach for Arabic Manuscripts Classification"

_sensors, 2023, doi:10.3390/s23198133_

Round 1
Reviewer 1 Report
The paper deals with important tasks. It has a scientific novelty and great practical value. It has a logical structure. The article is technically sound. The experimental section is good. The proposed approach is logical, results are clear.
Positive sides:
1. The article is well organized, the material is presented sequentially with a logical conclusion.
2. Develop a comprehensive framework (Frame-81) for processing historical Arabic manuscripts by utilizing catalog cards with parameter identification.
3. Employ the Faster R-CNN model to detect and categorize Arabic text as either typed or handwritten (manuscript).
4. Utilize the Faster R-CNN model for identifying and extracting parameter values from the catalog cards.
5. A discussion of the obtained results was made.
Suggestions:
1. Introduction section should be extended using more clearly the motivation of this paper.
2. It would be good to add the remainder of this paper.
3. I believe that augmenting the data with various image transformations can potentially improve the model's performance on complex manuscripts.
4. Given that some regions of interest (RoIs) were not accurately detected, I recommend considering the possibility of applying post-processing methods.
5. I also consider it appropriate to expand the introduction section by analyzing ensemble methods of digital image processing for handwriting recognition. In particular, pay attention to such publications doi: 10.1016/j.compeleceng.2021.107111, doi: 10.1109/ACPR.2017.5 and doi: 10.1007/978-3-031-24475-9_56.
6. Authors should provide a link to open access repository with the dataset in the Reference section.
Author Response
The paper deals with important tasks. It has a scientific novelty and great practical value. It has a logical structure. The article is technically sound. The experimental section is good. The proposed approach is logical, results are clear.
Positive sides:
- The article is well organized, the material is presented sequentially with a logical conclusion.
- Develop a comprehensive framework (Frame-81) for processing historical Arabic manuscripts by utilizing catalog cards with parameter identification.
- Employ the Faster R-CNN model to detect and categorize Arabic text as either typed or handwritten (manuscript).
- Utilize the Faster R-CNN model for identifying and extracting parameter values from the catalog cards.
- A discussion of the obtained results was made.
Suggestions:
- Introduction section should be extended using more clearly the motivation of this paper.
Response:
The introduction has been extended, and the following paragraph clarifies the motivation.
This paper proposes a generalized framework for processing catalog cards of ancient Arabic manuscripts. The proposed approach identifies the parameters and their values using a set of processing steps and a deep learning technique, specifically, the Faster-RCNN and LSTM. The extraction process is template and layout-free, making it suitable for various card formats in different libraries. As such, the contributions of this paper are as follows: 1) Build a complete framework for ancient Arabic manuscripts processing based on the catalog cards with parameter identification. The processing aims to classify the cards into known and unknown, a crucial step in extracting information from ancient Arabic manuscripts. 2) Using Faster R-CNN to locate and classify the Arabic text into typed and handwritten text. The deep learning architecture facilitates accurately spotting regions representing typed and written text. 3) Using Faster RCNN to recognize the parameters and parameters’ values in the cards. This recognition extracts parts of the significant information found in the catalog cards of the ancient Arabic manuscripts.
- It would be good to add the remainder of this paper.
Response:
All parts have been added.
- I believe that augmenting the data with various image transformations can potentially improve the model’s performance on complex manuscripts.
Response:
As indicated in the conclusion, pre-processing and augmentation will be part of the future work.
Future work will be dedicated to processing all parameters found in the catalog cards and extracting information from these parameter values. Notably, the results were obtained without implementing any preprocessing steps except for the resizing step. Therefore, in future work, we plan to incorporate preprocessing stages, including image enhancement and noise removal, as well as post-processing methods, such as context-based OCR improvement, to address instances where Regions of Interest (RoIs) were not accurately detected.
- Given that some regions of interest (RoIs) were not accurately detected, I recommend considering the possibility of applying post-processing methods.
Response:
Post-processing will be part of the future work, as indicated in the conclusion.
Future work will be dedicated to processing all parameters found in the catalog cards and extracting information from these parameter values. Notably, the results were obtained without implementing any preprocessing steps except for the resizing step. Therefore, in future work, we plan to incorporate preprocessing stages, including image enhancement and noise removal, as well as post-processing methods, such as context-based OCR improvement, to address instances where Regions of Interest (RoIs) were not accurately detected.
- I also consider it appropriate to expand the introduction section by analyzing ensemble methods of digital image processing for handwriting recognition. In particular, pay attention to such publications doi: 10.1016/j.compeleceng.2021.107111, doi: 10.1109/ACPR.2017.5 and doi: 10.1007/978-3-031-24475-9_56.
Response:
These papers and related papers were added to the discussion of the related work and the summary given in Table 1.
To ease the OCR process and enhance its performance in documents containing multiple lines, words, and regions, word-spotting techniques have been proposed. Word-spotting techniques are developed to locate and recognize specific words or phrases within images of documents. Word spotting is typically implemented as a pre-processing step before OCR to extract specific information among various information provided. Daraee, et al. [37] proposed a framework for word spotting for the applications of query-by-example and query-by-string. A Monte-Carlo dropout CNN is used for the word-spotting task. Monteiro, et al. [38], on the other hand, proposed a framework combining object detection using the YOLO technique, which uses CNN deep learning architecture, with OCR. This integrated framework allows for the recognition of both textual and non-textual objects, particularly in printed stickers. Jemni, et al. [39] proposed a keyword-spotting framework that relies on a transformer-based deep learning architecture, eliminating the need for CNN layers. This approach aims to generate more robust and semantically meaningful features, thereby enhancing keyword-spotting performance.
Table 1: Summary of the Related Work
|
Ref. |
App. |
Advantage |
Disadvantage |
|
[16] |
Arabic OCR |
Easy to interpret |
Low performance |
|
[17] |
Noise tolerance |
Not suitable for handwritten text |
|
|
[18] |
Fast and efficient |
Not suitable for distorted text, or noisy images |
|
|
[20,22,23] |
Noise tolerance |
Not suitable for handwritten text |
|
|
[26,27,29,31,34,35] |
No preprocessing |
Suits single character images only |
|
|
[37-39] |
Spotting |
No preprocessing |
Captured single word only |
- Authors should provide a link to open access repository with the dataset in the Reference section.
Response:
The collected dataset is part of the fund with privacy agreement, the dataset will be provided upon request after signing the agreement.
Reviewer 2 Report
-
The language usage throughout this paper need to be improved, the author should do some proofreading on it. Give the article a mild language revision to get rid of few complex sentences that hinder readability and eradicate typo errors.
-
Spell out each acronym the first time used in the body of the paper. Spell out acronyms in the Abstract.
-
What is the motivation of the proposed work? Research gaps, objectives of the proposed work should be clearly justified.
-
The authors should consider more recent research done in the field of their study (especially in the years 2022 and 2023 onwards).
-
Introduction needs to explain the main contributions of the work more clearly. The novelty of this paper is not clear. The difference between present work and previous Works should be highlighted. Introduction section can be extended to add the issues in the context of the existing work and how proposed algorithms/approach can be used to overcome this.
-
In the references, the authors cite some works. However, they have not indicated the advantage or disadvantage and their relations to this paper. It’s a little confusing.
-
Comparsion with recent study and methods would be appreciated.
-
Literature review techniques has to be strengthened by including the issues in the current system and how the author proposes to overcome the same.
-
The paper does not explain clearly its advantages with respect to the literature: it is not clear what is the novelty and contributions of the proposed work: does it propose a new method? Or does the novelty only consists in the application?
-
The advantage of the proposed method with respect to other methods in the literature should be clarified.
-
The paper does not provide significant experimental details needed to correctly assess its contribution: What is the validation procedure used?
-
Need detailed explanation of the preprocessing steps.
-
Clarify the finding Error rate and accuracy in performance analysis section.
-
Introduce the chart for given algorithm with description.
-
Please add information about the time to label a new sample under analysis. How does this work compare to other works? The contributions of this work need to be clearly articulated. The author might consider justifying the performance of this study with recent study and methods. What about the time to build the models? The method/approach in the context of proposed work should be written in detail.
-
Authors should add more details about the implementation of the code to perform the analysis and the library involved in this task.
-
Authors should add the parameters of the algorithms. What are the parameters used in the proposed system and how their values are set? Also, how the parameter values can affect the proposed system? Sections like Experimentation have to be extended and improved thus providing a more convincing contribution to the paper.
-
The authors provided details about the implementation setup and working environment. However, some training info should also be given in experimental section. How long does the proposed approach take to learn parameter? These details are missing and must be added to keep the paper standalone.
-
Authors have proven that their model is able to generate more acceptable, but has not indicated the practical application and effects of such a solution.
-
An error and statistical analysis of data should be performed.
-
A comparison with state of art (2022/2023) in the form of table should be added
-
More motivation/context regarding the application side of it, particularly on the aspects that make this technique particularly suited for industrial application scenarios, and how it would be applied in real scenarios. These aspects could additionally be supported with some related work context.
The authors "focused on neuroimage processing”, however, this decision is not clear. They must include a better explanation of the contribution to the introduction section, as well as explain about online monitoring, detection, and support of the diagnosis of diseases. To illustrate the contribution, I suggest adding a figure which shows technological advances throughout the last years.
“Machine and Deep Learning” has widely been developed in several research lines, and in this paper, only its applications of them on Neuroimage are mentioned. I propose improving each section in order to introduce the technologies and techniques in a better way.
Correlate it with other current Technologies, such as: IoT (communications, networks, Cloud, …), in terms of latency I guess that this field is quite sensitive to the delays required to process data, which should call for new investigations around the tradeoff between learning cost and performance (e.g. Deep Learning is costly, yet attains good predictive scores… should we opt for weak learners over good features? Or complex learners over raw data? Or a mixture of both of them, e.g. learned features off-line + weak learners on-line? Should data be sent to the cloud? Be preprocessed at the edge?). This issue is also very trendy at the communications level.
I would also suggest including aspects related to data such as ethics/explainable AI, which are of utmost importance in the medical domain. Also, the lack of annotated data of admissible quality is a problem for certain diseases, for which data augmentation techniques, transfer learning and domain adaptation are for sure fields within AI that the community should aim at in the near future.
# Associate Editor:
• The language usage throughout this paper need to be improved, the author should do some proofreading on it. Give the article a mild language revision to get rid of few complex sentences that hinder readability and eradicate typo errors.
• Overall, the basic background is not introduced well, where the notations are not illustrated much clear. I recommend the authors to employ certain intuitive examples to elaborate the essential notations.
• Spell out each acronym the first time used in the body of the paper. Spell out acronyms in the Abstract and title.
• The abstract can be rewritten to be more meaningful. The authors should add more details about their final results in the abstract. Abstract should clarify what is exactly proposed (the technical contribution) and how the proposed approach is validated.
• What is the motivation of the proposed work? Research gaps, objectives of the proposed work should be clearly justified.
• The authors should consider more recent research done in the field of their study (especially in the years 2022 and 2023 onwards).
• Introduction needs to explain the main contributions of the work more clearly. Theoretical contribution should be specified, if any.
• In the references in the Introduction section, the authors cite some works. However, they have not indicated the advantage or disadvantage and their relations to this paper. It’s a little confusing. Literature review techniques has to be strengthened by including the issues in the current system and how the author proposes to overcome the same.
• Comparsion with recent study (2022/2023) and methods would be appreciated.
• The paper does not explain clearly its advantages with respect to the literature: it is not clear what is the novelty and contributions of the proposed work: does it propose a new method? Or does the novelty only consists in the application?
• Results need more explanations. Additional analysis is required at each experiment to show the its main purpose.
• Need detailed explanation of the preprocessing steps.
• Clarify the finding Error rate and accuracy in performance analysis section.
• Introduce the chart for given algorithm with description.
• Please add information about the time to label a new sample under analysis.
• Authors should add more details about the implementation of the code to perform the analysis and the library involved in this task.
• Authors should add the parameters of the algorithms.
• What are the parameters used in the proposed system and how their values are set? Also, how the parameter values can affect the proposed system? Sections like Experimentation have to be extended and improved thus providing a more convincing contribution to the paper.
• The authors provided details about the implementation setup and working environment. However, some training info should also be given in experimental section. How long does the proposed approach take to learn parameter? These details are missing and must be added to keep the paper standalone.
• More motivation/context regarding the application side of it, particularly on the aspects that make this technique particularly suited for industrial application scenarios, and how it would be applied in real scenarios. These aspects could additionally be supported with some related work context.
• Correlate it with other current Technologies, such as: IoT (communications, networks, Cloud, …), in terms of latency I guess that this field is quite sensitive to the delays required to process data, which should call for new investigations around the tradeoff between learning cost and performance (e.g. Deep Learning is costly, yet attains good predictive scores… should we opt for weak learners over good features? Or complex learners over raw data? Or a mixture of both of them, e.g. learned features off-line + weak learners on-line? Should data be sent to the cloud? Be preprocessed at the edge?). This issue is also very trendy at the communications level.
• I would also suggest including aspects related to data such as ethics/explainable AI, which are of utmost importance in this domain. Also, the lack of annotated data of admissible quality is a problem for certain diseases, for which data augmentation techniques, transfer learning and domain adaptation are for sure fields within AI that the community should aim at in the near future.
Must be improved
Author Response
- The language usage throughout this paper need to be improved, the author should do some proofreading on it. Give the article a mild language revision to get rid of few complex sentences that hinder readability and eradicate typo errors.
Response:
A proofreading was conducted to improve the language of the paper. Here are the sentences that have been modified (some minor modification are highlighted in the main paper.).
Abstract:
The whole abstract has been re-written.
Abstract: For centuries, libraries worldwide have preserved ancient manuscripts due to their immense historical and cultural value. However, over time, both natural and human-made factors have led to the degradation of many of these ancient Arabic manuscripts, causing the loss of significant information such as authorship, titles, or subjects, rendering them as unknown manuscripts. Although catalog cards attached to these manuscripts might contain some of the missing details, these cards have degraded significantly in quality over the decades within libraries. This paper presents a framework for identifying these unknown ancient Arabic manuscripts by processing the catalog cards associated with them. Given the challenges posed by the degradation of these cards, simple optical character recognition (OCR) is often insufficient. The proposed framework uses deep learning architecture to identify unknown manuscripts within a collection of ancient Arabic documents. This involves locating, extracting, and classifying the text from these catalog cards, along with implementing processes for region-of-interest identification, rotation correction, feature extraction, and classification. The results demonstrate the effectiveness of the proposed method, achieving an accuracy rate of 92.5%, compared to 83.5% with classical image classification and 81.5% with OCR alone.
Introduction:
Line 44-56
Catalog cards contain information about books and manuscripts maintained in libraries. For the ancient manuscripts, these cards represent diligent work of librarians and researchers in identifying these invaluable resources, facilitating easy access to the information enclosed within these resources. Catalog cards for Arabic manuscripts typically include essential parameters, such as the title, the author’s name, the manuscript’s beginning and end, and the number of images [4]. Additional technical parameters like the number of lines, number of words, page size, paper type, and other details may be inferred, but they are not considered essential. These technical parameters serve various purposes, such as identifying the era in which the manuscript was written, reconstructing fragmented manuscripts, determining page layouts, and even discerning the author's style in cases where the author's name is unknown. Additionally, they can provide insights into the scribe's identity and the copying date when such information is missing [8].
Line 62-76
To extract information from documents, deep learning is employed. Deep learning algorithms performed well in the image-processing domain, including character and text recognition. Conventional Neural Networks (CNNs) are specifically used for image processing and computer vision tasks. While they perform well in Optical Character Recognition (OCR) of typed text, they struggle to capture multiple objects within the image. The main drawback of CNNs is their approach of processing the entire image as a single object, making it challenging to cannot be captured using CNN. To address this limitation, various techniques such as Region-based CNN (RCNN), Fast-RCNN, Faster-RCNN, and Mask RCNN have been proposed. Faster-RCNN, in particular, is a deep neural network algorithm that efficiently detects and recognizes objects in the image compared to RCNN. This algorithm can be efficiently used with proper preprocessing steps with complex OCR applications due to its high performance in complex object detection and recognition tasks [9]. On the other hand, Long Short-Term Memory (LSTM) networks are well-suited for learning patterns from sequence of data, making them a suitable choice for OCR applications [10].
Literature Review:
Line 103-105
Generally, manuscript processing requires image enhancement and segmentation into words and characters.
Line 110-118
The literature on ancient and historical manuscript processing is not as extensive as that on modern document processing. However, processing historical documents can benefit from the advances in modern document processing. Yet, ancient and historical manuscript processing poses more challenges due to the poor quality of the images [2,13-15]. Moreover, tasks associated with the ancient manuscripts are not typically required for modern documents, such as author identification and manuscript dating. There applications have not been extensively investigated in the literature [15]. Nevertheless, many of these tasks depend mainly on OCR-related applications with specific preprocessing or post-processing stages.
Line 126-127
The problem is converted into character recognition as the sliding window is applied.
Line 142-147
Recent trends in OCR techniques development have shifted towards segmentation-free approaches. These advancements have revealed that the use of segmentation techniques often leads to significant or trivial errors that can degrade OCR performance. In segmentation-free techniques, words are recognized without the need for prior segmentation step. For instance, Nemouchi, Meslati [13] proposed an OCR approach to recognizing handwritten city names.
Methodology and Design:
Line 260-261
The author’s name is the subject matter for the proposed approach, yet other information can be extracted similarly.
Line 264-265
The proposed framework identifies relevant text regions on the card among all other regions. Then, it classifies RoIs as typed and handwritten.
Line 274-275
The alternative framework experimented in the preliminary face of the research shares common characteristics with the proposed framework but uses the OCR, as illustrated in Figure 3.
Experiments:
Line 384-387
In this context, the typed text represented the parameters, and the handwritten text represents the values assigned to these parameters. Consequently, if the manuscript's parameter values were either not provided or were denoted as 'unknown', the manuscript was classified as 'unknown'.
Line 387-388
On the contrary, it is classified as known if the parameters are complete.
Results and Analysis:
Line 441-442
As given, the proposed framework outperformed the baselines and the alternative frameworks regarding accuracy, precision, recall, and f-measure.
Line 447
As for the written text, for 2105 regions, 1979 regions were recognized.
Conclusion and Future Work
Line 473-780
This paper proposed a framework for classifying ancient Arabic manuscripts into known and unknown categories. The framework primary focus is identifying parameters and recognizing their values from catalog cards associated with the manuscripts. The framework processes the input cards through several steps, including a region detection phase, rotation correction, and region alignments, followed by the final classification step. For this purpose, Faster RCNN and various classification approaches were utilized. A dataset was collected from various libraries and annotated with bounding boxes corresponding to the objects of interest.
- Spell out each acronym the first time used in the body of the paper. Spell out acronyms in the Abstract.
Response:
The acronyms were spell out as it appears in the first time. Here are the acronyms that were modified.
Abstract:
Line 23
The results showed that the proposed method achieved an accuracy of 92.5% compared to using classical image classification, which achieved 83.5%, and solely Optical Character Recognition (OCR), which achieved 81.5%.
Introduction:
Line 70
Although it performs well with Optical Character Recognition (OCR) of typed text, it fails to capture repeated patterns in sequential and temporal data.
Line 73-75
Accordingly, Region-based CNN (RCNN), Fast-RCNN, Faster-RCNN, and Mask RCNN were proposed. The Faster-RCNN is a deep neural network algorithm that detects and recognizes objects in the image more efficiently than the RCNN.
Literature Review:
Line 117
The experiments were conducted on the International Conference on Document Analysis and Recognition (ICDAR)-2007 competitions dataset for word recognition, with an accuracy of 94.85%.
Line 121-122
Applying this technique to the ‘Institut fur Nachrichtentechnik’ and ‘Ecole Nationale d’Ingénieurs de Tunis’ (IFN/ENIT) dataset [19], which involves 32492 handwritten words, achieved an accuracy of 95.15%.
Line 178
The experiments are conducted on the Arabic Handwriting Data Base (AHDB) [27] dataset of 4971 samples of 50 classes of single literal amounts with 100 samples each.
Line 184-185
The experiments are conducted on a National Institute of Standards and Technology (NIST) special dataset of digits, called Modified NIST (MNIST) [29], consisting of handwritten images of 240000 samples.
Line 193-195
The experiments were conducted on AlexU Isolated Alphabet (AIA9K) [32], and Arabic Handwritten Character Dataset (AHCD) [16] datasets; the results obtained for these datasets were 94.8% and 97.6%, respectively.
Line 198-200
The results for the proposed technique over the Center for Microprocessor Applications for Training Education and Research (CMATER) dataset were 99.4%.
Methodology and Design
Line 315
The preprocessed input images undergo a recognition process that implements LSTM, another deep learning approach based on Recurrent Neural Network (RNN), which can be used to process data sequences, such as image pixels.
Experiments:
Line 355-357
Linear Support Vector Machine (SVM), KNN (k=3), Decision Tree (DT), Random Forest (RF), Artificial Neural Network (ANN), and Deep Neural Network (Deep-NN).
Line 381-383
The proposed framework was implemented using Python in the Anaconda Integrated Development Environment (IDE) and based on a set of libraries as given in Table 1.
- What is the motivation of the proposed work? Research gaps, objectives of the proposed work should be clearly justified.
Response:
The motivation, research gap and objectives have been added to the manuscripts and the introduction have been rearranged to fit the context.
This paper proposes a generalized framework for processing catalog cards of ancient Arabic manuscripts. The proposed approach identifies the parameters and their values using a set of processing steps and a deep learning technique, specifically, the Faster-RCNN and LSTM. The extraction process is template and layout-free, making it suitable for various card formats in different libraries. As such, the contributions of this paper are as follows: 1) Build a complete framework for ancient Arabic manuscripts processing based on the catalog cards with parameter identification. The processing aims to classify the cards into known and unknown, a crucial step in extracting information from ancient Arabic manuscripts. 2) Using Faster R-CNN to locate and classify the Arabic text into typed and handwritten text. The deep learning architecture facilitates accurately spotting regions representing typed and written text. 3) Using Faster RCNN to recognize the parameters and parameters’ values in the cards. This recognition extracts parts of the significant information found in the catalog cards of the ancient Arabic manuscripts.
- The authors should consider more recent research done in the field of their study (especially in the years 2022 and 2023 onwards).
Response:
Recent Studies was added to the related work in the following paragraph.
To ease the OCR process and enhance its performance in documents containing multiple lines, words, and regions, word-spotting techniques have been proposed. Word-spotting techniques are developed to locate and recognize specific words or phrases within images of documents. Word spotting is typically implemented as a pre-processing step before OCR to extract specific information among various information provided. Daraee, et al. [37] proposed a framework for word spotting for the applications of query-by-example and query-by-string. A Monte-Carlo dropout CNN is used for the word-spotting task. Monteiro, et al. [38], on the other hand, proposed a framework combining object detection using the YOLO technique, which uses CNN deep learning architecture, with OCR. This integrated framework allows for the recognition of both textual and non-textual objects, particularly in printed stickers. Jemni, et al. [39] proposed a keyword-spotting framework that relies on a transformer-based deep learning architecture, eliminating the need for CNN layers. This approach aims to generate more robust and semantically meaningful features, thereby enhancing keyword-spotting performance.
- Introduction needs to explain the main contributions of the work more clearly. The novelty of this paper is not clear. The difference between present work and previous Works should be highlighted. Introduction section can be extended to add the issues in the context of the existing work and how proposed algorithms/approach can be used to overcome this.
Response:
The contribution in the last paragraph of the introduction was elaborated.
This paper proposes a generalized framework for processing catalog cards of ancient Arabic manuscripts. The proposed approach identifies the parameters and their values using a set of processing steps and a deep learning technique, specifically, the Faster-RCNN and LSTM. The extraction process is template and layout-free, making it suitable for various card formats in different libraries. As such, the contributions of this paper are as follows: 1) Build a complete framework for ancient Arabic manuscripts processing based on the catalog cards with parameter identification. The processing aims to classify the cards into known and unknown, a crucial step in extracting information from ancient Arabic manuscripts. 2) Using Faster R-CNN to locate and classify the Arabic text into typed and handwritten text. The deep learning architecture facilitates accurately spotting regions representing typed and written text. 3) Using Faster RCNN to recognize the parameters and parameters’ values in the cards. This recognition extracts parts of the significant information found in the catalog cards of the ancient Arabic manuscripts.
- Comparison with recent study and methods would be appreciated.
Response:
Comparison was add by summarizing the related work and distinguish them from the proposed work as given in the last section of the related work.
A summary of the related work is given in Table 1. In summary, advanced deep learning algorithms are typically designed for recognizing single entities, such as characters, symbols, numbers, words, or phrases. Traditional OCR systems often struggle with hand-written text. To address these limitations, this paper proposes a framework for recognizing multiple entities to facilitate information extraction.
Table 1: Summary of the Related Work
|
Ref. |
App. |
Advantage |
Disadvantage |
|
[16] |
Arabic OCR |
Easy to interpret |
Low performance |
|
[17] |
Noise tolerance |
Not suitable for handwritten text |
|
|
[18] |
Fast and efficient |
Not suitable for distorted text, or noisy images |
|
|
[20,22,23] |
Noise tolerance |
Not suitable for handwritten text |
|
|
[26,27,29,31,34,35] |
No preprocessing |
Suits single character images only |
|
|
[37-39] |
Spotting |
No preprocessing |
Captured single word only |
- In the references, the authors cite some works. However, they have not indicated the advantage or disadvantage and their relations to this paper. It’s a little confusing.
Response:
Table 1 with advantages and disadvantages was added.
Table 1: Summary of the Related Work
|
Ref. |
App. |
Advantage |
Disadvantage |
|
[16] |
Arabic OCR |
Easy to interpret |
Low performance |
|
[17] |
Noise tolerance |
Not suitable for handwritten text |
|
|
[18] |
Fast and efficient |
Not suitable for distorted text, or noisy images |
|
|
[20,22,23] |
Noise tolerance |
Not suitable for handwritten text |
|
|
[26,27,29,31,34,35] |
No preprocessing |
Suits single character images only |
|
|
[37-39] |
Spotting |
No preprocessing |
Captured single word only |
- Literature review techniques has to be strengthened by including the issues in the current system and how the author proposes to overcome the same.
Response:
Table 1 with advantages and disadvantages was added.
- The paper does not explain clearly its advantages with respect to the literature: it is not clear what is the novelty and contributions of the proposed work: does it propose a new method? Or does the novelty only consists in the application?
Response:
The novelty of the proposed framework was added.
A summary of the related work is given in Table 1. In summary, advanced deep learning algorithms are typically designed for recognizing single entities, such as characters, symbols, numbers, words, or phrases. Traditional OCR systems often struggle with hand-written text. To address these limitations, this paper proposes a framework for recognizing multiple entities to facilitate information extraction.
- The advantage of the proposed method with respect to other methods in the literature should be clarified.
Response:
The advantages of the proposed framework was added.
A summary of the related work is given in Table 1. In summary, advanced deep learning algorithms are typically designed for recognizing single entities, such as characters, symbols, numbers, words, or phrases. Traditional OCR systems often struggle with hand-written text. To address these limitations, this paper proposes a framework for recognizing multiple entities to facilitate information extraction.
- The paper does not provide significant experimental details needed to correctly assess its contribution: What is the validation procedure used?
Response:
No extra validation is required. The classification outcomes is obtained based on the validation set.
- Need detailed explanation of the preprocessing steps.
Response:
No preprocessing is implemented. This is will be part of the future work as given in the conclusion section.
Future work will be dedicated to processing all parameters found in the catalog cards and extracting information from these parameter values. Notably, the results were obtained without implementing any preprocessing steps except for the resizing step. Therefore, in future work, we plan to incorporate preprocessing stages, including image enhancement and noise removal, as well as post-processing methods, such as context-based OCR improvement, to address instances where Regions of Interest (RoIs) were not accurately detected.
- Clarify the finding Error rate and accuracy in performance analysis section.
Response:
Discussion were provided in the results section.
- Introduce the chart for given algorithm with description.
Response:
The flow chart is exists (Figure 2, Figure 3, Figure 4, Figure 5, and Figure 6)
- Please add information about the time to label a new sample under analysis. How does this work compare to other works? The contributions of this work need to be clearly articulated. The author might consider justifying the performance of this study with recent study and methods. What about the time to build the models? The method/approach in the context of proposed work should be written in detail.
Response:
Time of the deep learning was discussed.
- Authors should add more details about the implementation of the code to perform the analysis and the library involved in this task.
Response:
Libraries are presented and discussed in Table 2.
Table 2: List of Utilized Libraries
|
Library |
Description |
|
Numpy |
Used for array manipulation, specifically for saving the titles and the information of the images and the output of region detection and to sort the regions based on the coordinates. |
|
Scikit-Learn |
Used for the classification step to implement the utilized classifiers. |
|
OpenCV |
Used for image reading, writing, and in the preprocessing stages with the OCR. |
|
Tesseract 4.0 |
Used as the OCR engine based on LSTM with a pre-trained Arabic model. |
- Authors should add the parameters of the algorithms. What are the parameters used in the proposed system and how their values are set? Also, how the parameter values can affect the proposed system? Sections like Experimentation have to be extended and improved thus providing a more convincing contribution to the paper.
Response:
Parameters are presented and discussed in the results and analysis.
- The authors provided details about the implementation setup and working environment. However, some training info should also be given in experimental section. How long does the proposed approach take to learn parameter? These details are missing and must be added to keep the paper standalone.
Response:
Experimental details are provided in the results and analysis.
- Authors have proven that their model is able to generate more acceptable, but has not indicated the practical application and effects of such a solution.
Response:
The application was discussed in the Introduction (Identifying unknown manuscripts automatically)
- An error and statistical analysis of data should be performed.
Response:
The results are presented using the statistical measures accuracy, precision, recall and F-Measure.
- A comparison with state of art (2022/2023) in the form of table should be added.
Response:
Recent Studies will be added.
To ease the OCR process and enhance its performance in documents containing multiple lines, words, and regions, word-spotting techniques have been proposed. Word-spotting techniques are developed to locate and recognize specific words or phrases within images of documents. Word spotting is typically implemented as a pre-processing step before OCR to extract specific information among various information provided. Daraee, et al. [37] proposed a framework for word spotting for the applications of query-by-example and query-by-string. A Monte-Carlo dropout CNN is used for the word-spotting task. Monteiro, et al. [38], on the other hand, proposed a framework combining object detection using the YOLO technique, which uses CNN deep learning architecture, with OCR. This integrated framework allows for the recognition of both textual and non-textual objects, particularly in printed stickers. Jemni, et al. [39] proposed a keyword-spotting framework that relies on a transformer-based deep learning architecture, eliminating the need for CNN layers. This approach aims to generate more robust and semantically meaningful features, thereby enhancing keyword-spotting performance.
- More motivation/context regarding the application side of it, particularly on the aspects that make this technique particularly suited for industrial application scenarios, and how it would be applied in real scenarios. These aspects could additionally be supported with some related work context.
Response:
The motivation, research gap and objectives have been added to the manuscripts and the introduction have been rearranged to fit the context.
This paper proposes a generalized framework for processing catalog cards of ancient Arabic manuscripts. The proposed approach identifies the parameters and their values using a set of processing steps and a deep learning technique, specifically, the Faster-RCNN and LSTM. The extraction process is template and layout-free, making it suitable for various card formats in different libraries. As such, the contributions of this paper are as follows: 1) Build a complete framework for ancient Arabic manuscripts processing based on the catalog cards with parameter identification. The processing aims to classify the cards into known and unknown, a crucial step in extracting information from ancient Arabic manuscripts. 2) Using Faster R-CNN to locate and classify the Arabic text into typed and handwritten text. The deep learning architecture facilitates accurately spotting regions representing typed and written text. 3) Using Faster RCNN to recognize the parameters and parameters’ values in the cards. This recognition extracts parts of the significant information found in the catalog cards of the ancient Arabic manuscripts.
- The authors "focused on neuroimage processing”, however, this decision is not clear. They must include a better explanation of the contribution to the introduction section, as well as explain about online monitoring, detection, and support of the diagnosis of diseases. To illustrate the contribution, I suggest adding a figure which shows technological advances throughout the last years.
“Machine and Deep Learning” has widely been developed in several research lines, and in this paper, only its applications of them on Neuroimage are mentioned. I propose improving each section in order to introduce the technologies and techniques in a better way.
Correlate it with other current Technologies, such as: IoT (communications, networks, Cloud, …), in terms of latency I guess that this field is quite sensitive to the delays required to process data, which should call for new investigations around the tradeoff between learning cost and performance (e.g. Deep Learning is costly, yet attains good predictive scores… should we opt for weak learners over good features? Or complex learners over raw data? Or a mixture of both of them, e.g. learned features off-line + weak learners on-line? Should data be sent to the cloud? Be preprocessed at the edge?). This issue is also very trendy at the communications level.
I would also suggest including aspects related to data such as ethics/explainable AI, which are of utmost importance in the medical domain. Also, the lack of annotated data of admissible quality is a problem for certain diseases, for which data augmentation techniques, transfer learning and domain adaptation are for sure fields within AI that the community should aim at in the near future.
Response:
The comments provided here most properly are by mistakes. There is neuroimaging, weak-learners, ethics or diseases in the paper

Reviewer 3 Report
The article's strength lies in its comprehensive yet succinct literature review, and in the detailed explanation of the deep learning architecture employed to recognize obscure manuscripts in the collection of ancient Arabic scripts. The outcomes presented are convincing. However, the "Conclusions and Future Work" could benefit from a slightly more detailed account of the results.
Author Response
The article's strength lies in its comprehensive yet succinct literature review, and in the detailed explanation of the deep learning architecture employed to recognize obscure manuscripts in the collection of ancient Arabic scripts. The outcomes presented are convincing. However, the "Conclusions and Future Work" could benefit from a slightly more detailed account of the results.
Response:
The conclusion were extended as given.
This paper proposed a framework for classifying ancient Arabic manuscripts into known and unknown categories. The framework primary focus is identifying parameters and recognizing their values from catalog cards associated with the manuscripts. The framework processes the input cards through several steps, including a region detection phase, rotation correction, and region alignments, followed by the final classification step. For this purpose, Faster RCNN and various classification approaches were utilized. A dataset was collected from various libraries and annotated with bounding boxes corresponding to the objects of interest. The experiments conducted on this dataset showed that the proposed approach outperformed the classical approaches by 10% in terms of accuracy. The results showed that the proposed framework is very robust detection and recognition tasks. The Faster RCNN achieved full results distinguishing the typed text from the written text. However, some RoI were not recognized, especially for the written text. Accordingly, for 3200 regions for typed text, 3104 were recognized fully or partially. As for the written text, for 2105 regions, 1979 regions were recognized.
Future work will be dedicated to processing all parameters found in the catalog cards and extracting information from these parameter values. Notably, the results were obtained without implementing any preprocessing steps except for the resizing step. Therefore, in future work, we plan to incorporate preprocessing stages, including image enhancement and noise removal, as well as post-processing methods, such as context-based OCR improvement, to address instances where Regions of Interest (RoIs) were not accurately detected.

Round 2
Reviewer 2 Report
The paper deserves to be accepted
Good